# High Non-Cardiac Death Incidence Should Be a Limitation of Drug-Eluting Stents Implantation? Insights from Recent Randomized Data

**DOI:** 10.3390/diagnostics13071321

**Published:** 2023-04-02

**Authors:** Alfredo E. Rodriguez, Carlos Fernandez-Pereira, Juan Ramon Mieres, Alfredo Matias Rodriguez-Granillo

**Affiliations:** Cardiac Unit, Otamendi Hospital, Post Graduate Buenos Aires School of Medicine, Cardiovascular Research Center (CECI), 868 Buenos Aires, Argentina

**Keywords:** percutaneous coronary interventions, stents, drug-eluting stents, coronary artery bypass surgery, revascularization therapies, non-cardiac death, optimal medical treatment

## Abstract

Despite the introduction of drug-eluting stents (DES) significantly improved the efficacy and safety of percutaneous coronary interventions (PCI), particularly in a high-risk group of patients, the gap between PCI with his competitor’s coronary artery bypass surgery (CABG) and/or optimal medical treatment alone was not reduced. In this revision, we highlighted the fact that in recent years landmark randomized studies reported at mid and long-term follow-ups a high incidence of non-cardiac death, cancer incidence, or both in the DES group of patients. The overall incidence of non-cardiac death was significantly higher in the DES vs. the comparator arm: 5.5% and 3.8%, respectively, *p* = 0.000018, and non-cardiac death appears to be more divergent between DES vs. the comparator at the extended follow-up to expenses of the last one. One of these trials reported five times greater cancer incidence in the DES arm at late follow-up, 5% vs. 0.7% *p* < 0.0018. We review the potential reason for these unexpected findings, although we can discard that DES biology could be involved in it. Until all these issues are resolved, we propose that DES implantation should be tailored accorded patient age, life expectancy, and lesion complexity.

## 1. Introduction

Percutaneous coronary intervention (PCI) with drug-eluting stents (DES) has become the most effective tool to reduce angiographic coronary restenosis (ISR), repeat revascularization procedures in the target vessel (TVR), and stent thrombosis after PCI when compared to bare metal stents (BMS), although, all cause of death was not changed [1] and the benefits of DES compared to BMS was especially seen in the left anterior descending coronary artery [2].

Despite the above advantages, DES has been unable to reduce the gap of PCI either compared to coronary artery bypass surgery (CABG) or optimal medical treatment (OMT) in terms of serious adverse events such as all causes of death or myocardial infarction (MI) [3,4,5,6,7].

During the last 17 years, we made the hypothesis that the complexity of the lesion subsets treated in the DES era, unprotected left main coronary artery stenosis (LMCA), and complex three vessels coronary artery disease (CAD) could be the reason for these negative findings. However, the lack of advantage over the competitors remained even in a more CAD-favorable scenario [8,9]; in fact, we have reported concerns related to late cardiac adverse events, including a high incidence of MI after DES implantation [10]. However, to our knowledge, only isolated data from an old meta-analysis [11] highlighted the potential link between DES use and the incidence of high non-cardiac death.

It was only recently that major landmarks in randomized clinical trials (RCT) reported at long-term follow-up either an unexpected increase in non-cardiac death or cancer incidence in the DES arm and these worrisome findings are the purposes of this Comprehensive revision.

In the following pages, we will first briefly describe all trials reporting a high incidence of non-cardiac death and/or cancer occurrence in the last 6 years.

In the second section, we will analyze a few previous similar data published in the literature when 1st DES designs were introduced in clinical practice.

And finally, we hypothesize the possible reasons for these findings finishing this revision with conclusions and perspective.

## 2. Increased Incidence of Non-Cardiac Adverse Events in Recent Randomized Clinical Trials

In the last 5 years, RCT reported an unexpectedly high incidence of non-cardiac and/or cancer incidence in the DES compared to the control arm (Table 1 and Figure 1 and Figure 2), with only randomized trials reporting a higher incidence of non-cardiac death against their comparator and having 3 or more years of follow-up was selected.

As described in Table 1, follow-up length was 5 years in BIOSCIENCE (2119 patients) and EXCEL (1905 patients), 5.7 years in ISCHEMIA EXTENDED (4825 patients), 3.4 years in REVIVED (700 patients), and 3.2 years (5179 patients) in ISCHEMIA.

## 3. Trials List (Table 1)

1.In 2018, a BIOSCIENCE trial (3) compared a new biodegradable sirolimus thin stent struts design vs. an everolimus durable polymer design. The trial was an initial investigator, single-blind, multicenter, randomized, non-inferiority trial. Patients were followed up at 1 month, 1 year, 2 years, and 5 years by telephone interview or clinical visit.

It showed at 5 years a significantly high incidence of all-cause of death in the biodegradable sirolimus thin stent struts design (14.1% vs. 10.3% respectively; *p* = 0.017). This difference was driven by a higher non-cardiac death incidence in the sirolimus thin stent struts, with a two times increase in the incidence of cancer death (2.7% vs. 1.3% *p* = 0.037) [3]. The difference in all-cause mortality emerged after 1 year of stent deployment.

Despite the authors of the article declaring their interest in finding out the reasons for that, to our knowledge, no further comments about this issue were published by the authors or study sponsors, and this stent design is routinely used in clinical practice all over the world. In addition, a large meta-analysis of 16 trials, 20,707 patients with 2.5 years of follow-up, also demonstrates a non-significant but numerically greater all-cause mortality with ultrathin-strut DES compared to 2nd generation DES (1.11,098–1.26) [12].

Furthermore, recently a mid-sample size trial with this DES published 5 years follow-up results [11], and it did not show an increase in non-cardiac death with this stent; however, if we pooled the data of these two studies, non-cardiac death is still significantly higher with ultrathin stent struts compared to everolimus-eluting stents (5.2% vs. 3.4%, respectively, *p* < 0.021) [13].

2.In 2019 [4], the largest RCT between PCI and CABG in non-protective LMCA, EXCEL (Evaluation of XIENCE versus Coronary Artery Bypass Surgery for Effectiveness of Left Main Revascularization) trial, published 5 years of follow-up results.

The primary endpoint was the composite of any cause of death, cerebrovascular accident (CVA), or MI at 3 years. Long-term additional secondary outcomes included all of these and their individual components at 5 years. The primary end point of death, CVA, or MI at 5 years occurred in 22.0% and 19.2% in the PCI and CABG groups, respectively (*p* = 0.13). However, all-cause of death was significantly in favor of CABG due to an unexpectedly high incidence of non-cardiac death, and the difference occurred between 1st to 5th year of follow-up (10% vs. 6.6% with PCI and CABG, respectively, HR 1.57, 1.12–2.19).

Despite these significant late mortality differences, the authors suggested that this finding may be due to chance, and no other comments were made further.

3.In 2020, the largest RCT between an invasive strategy (PCI and CABG 74% and 26%, respectively) and OMT, ISCHEMIA (International Study of Comparative Health Effectiveness with Medical and Invasive Approaches) trial reported a 3.2-year follow-up outcome of the 5179 patients enrolled in the trial [5]. The primary endpoint was a composite of cardiovascular death, MI, or hospitalization for unstable angina, heart failure, or resuscitated cardiac arrest. Secondary outcomes were the composite of cardiovascular death or MI and angina-related quality of life. The primary endpoint occurred in 318 patients in the invasive-strategy group and in 352 patients in the conservative-strategy group. In an adjusted Cox model analysis, the estimated HR with the invasive strategy as compared with the conservative strategy was 0.93 (95%CI, 0.80 to 1.08; *p* = 0.34).

However, despite of main results of this study showed no significant differences in primary outcomes, overall or cardiac death between both revascularization strategies, non-cardiac death was higher in the invasive compared to the conservative group (2.0% vs. 1.3%, respectively, *p* = 0.029) mostly due to cancer. At the time the study was presented, there was a trend to lower cardiac mortality in the invasive group of patients [5,8,14].

4.In the first quartile of 2022, REVIVED (Revascularization for Ischemic Ventricular Dysfunction) trial, a randomized comparison between DES vs. OMT in patients with ejection fraction ≤ 35%, was published [7]. Patients were eligible if they had a left ventricular ejection fraction of 35% or less, a severe jeopardized coronary artery score with demonstrable viability in at least four dysfunctional myocardial segments, and also, they should be amenable to revascularization with PCI. Patients with an acute MI in the last month before randomization were excluded. The primary composite endpoint was overall death or hospitalization for heart failure over a minimum follow-up period of 24 months.

The end point of death from any cause or hospitalization for heart failure was 37.2% in the PCI group and 38.0% in the OMT group (HR, 0.99; 95% CI, 0.78 to 1.27).

Therefore, the major result of this study was no mortality difference between DES and OMT.

However, if we look numbers of cardiovascular events, cardiac death and spontaneous MI were numerically and significantly lower, respectively, in the PCI Group: 21.9% vs. 24.9% and 5.2% vs. 9.3%, *p* = 0.03. In contrast, non-cardiac death was numerically higher with PCI than OMT (9.8% vs. 7.6%, respectively), including significant cancer incidence in the PCI group (5% vs. 0.7%, *p* = 0.0018); interestingly, authors reported this finding in the last table of the Appendix A of the study, but they did not highlight the five times increase of cancer in the main manuscript.

5.In November 2022, the authors of the ISCHEMIA trial reported 5.7 years of follow-up of 4825 from the 5125 patients initially included, now called ISCHEMIA EXTENDED, and their finding was no mortality advantage between PCI and OMT [7] despite cardiac death becoming now significantly lower with PCI, HR 0.78, 0.63, and 0.96.

In contrast, non-cardiac mortality was significantly higher with DES compared to the medical treatment arm, and the non-cardiac death survival curves are more divergent in favor of OMT over time (HR1.44,1.08, 1.91). Authors estimated an incidence of non-cardiac death at 7 years, 1.65% higher in the invasive strategy.

It seems to be PCI performed good work in terms of cardiac events, but for unknown reasons, non-cardiac adverse events equalized the final mortality curve.

Of interest, at 5.7 years, the non-cardiac mortality benefit of OMT compared to DES appears to be greater, but not significant, in patients with multiple vessel CAD (HR 1.52 0.88, 2.63) without any differences in a single vessel.

High non-cardiac death in more complex CAD features had been reported previously in Credo/Kyoto cohort 3 registry, where patients with high Syntax Scores treated with DES have greater non-cardiac mortality in comparison with CABG in the overall and after-matched population [15].

ISCHEMIA authors well described and analyzed this unexpected event in the discussion section of the main manuscript and expressed their concerns with this surprising finding.

In summary, in these RCTs briefly described above, BIOSCIENCE, EXCEL, and ISCHEMIA EXTENDED showed a significantly high incidence of non-cardiac death.

In BIOSCIENCE and ISCHEMIA EXTENDED, non-cardiac death was driven by a high incidence of cancer death, whereas in REVIVED with a lower length of follow-up than the other trials, a significantly high incidence of cancer occurrence was seen in the DES arm.

Of interest, this high cancer incidence in the DES group of patients observed in REVIVED would predict high late mortality in the DES-treated arm and a lack of benefit in the overall mortality rate of this trial.

Therefore, the results of ISCHEMIA EXTENDED and REVIVED trials reinforce our previous concerns related to the late increase incidence of non-cardiac adverse events observed in DES trials [10].

Many queries came up after these unpredicted results

Is this observation new?Is it only restricted to DES design?Are these findings related to prolonged DAPT therapy?Are these linked to DES biology?Are these findings connected to radiation?What is the role of endothelial dysfunction?Is endothelial dysfunction connected with tumors?Is endothelial dysfunction more prominent with multiple DES implantation?Is non-cardiac death occurred by chance?

We may have comments for all these queries, but without statements, the true answer is completely unknown today.

## 4. Previous Old Safety Data

The high incidence of non-cardiac death with DES design is not new [11]; it was reported previously and was never seen in BMS vs. CABG or OMT trials [16,17,18,19,20,21].

The one performed by Nordmann et al., 2006, [11], from 17 RCT, compared first-generation DES vs. BMS, and he found that trials using sirolimus-eluting stents were associated with a high incidence of non-cardiac death, including cancer in comparison with BMS. In the trial-level meta-analysis performed by Gaudino et al., 2020, [16] non-cardiac-related death was significantly higher compared to CABG in trials using DES (RR 1.28, 104–1.57) but not in those using BMS (RR 1.02, 0.75–1.38).

In fact, patient-level meta-analysis between PCI, either with balloon angioplasty or BMS versus CABG, did not observe differences in death, cardiac or non-cardiac death in the long-term, excluding patients with diabetes. Hlatky et al. and Daemen et al. both reported in 2008 and 2009 [18,19], respectively, comparative safety results of the two revascularization strategies at 5 years of follow-up, and that included patients with 2, 3 vessel disease and left anterior descending artery; therefore, in such analysis, the extension of coronary artery disease was not in favor to better survival with CABG. Moreover, in a sub-analysis performed by the same authors, Flather et al. in 2012 [20] observed a survival advantage with PCI in younger patients. Over a median follow-up of 5.9 years, the effect of CABG versus PCI on mortality varied according to age (interaction *p* < 0.01), with adjusted CABG-to-PCI HR of 1.23 (95% CI: 0.95 to 1.59) in the youngest tertile; 0.89 (95% CI: 0.73 to 1.10) in the middle tertile; and 0.79 (95% CI: 0.67 to 0.94) in the oldest tertile. A similar interaction of age with treatment was present for the composite outcome of death and MI. Therefore, in the pre-DES era, patient age modifies the comparative effectiveness of CABG and PCI on hard cardiac events, with CABG favored at older ages and PCI favored at younger ages. Thus, this analysis seems to be in the opposite direction that we recommended due to the late attrition rate observed in almost all RCTs comparing DES with CABG. [4,10,22,23,24,25,26,27,28,29]

In agreement, both metanalyses described in Table 1 reported a higher incidence of non-cardiac death with DES.

A high incidence of non-cardiac-related death was also seen in observational studies and registries in the past after DES implantation at long-term follow-up.

In fact, a reviewing article was written by us after a large observational study from New York was published in 2012 [25,26]; in such revision, we highlighted changes in the safety paradigm compared to CABG after the introduction of DES during PCI. However, at that time, we did not express concerns about non-cardiac-related death, and we were mainly focused on the role of late DES thrombosis as the main reason for those findings [30,31,32].

Weintraub et al. reported in a large observational registry from New York [25] a higher mortality rate with PCI in patients ≥65 years old in comparison with CABG, and the advantages of surgery were seen in all subgroups, including those at low risk. Despite the non-randomized nature, its results are largely strengthened by its sample size. The study included two prospective registries from 64 centers in the USA, and around 190,000 patients were included, matched in over 86,000; the differences in favor of CABG are significant overall and matched population and survival advantages with CABG remained after authors adjusted for clinical and angiographic variables.

In the ERACI III registry (Estudio Randomizado Argentino Angioplastia versus Cirugia) [30,31,32], we also found an increased incidence of death with DES beyond the first year in comparison with BMS or CABG groups; all causes of death at one year have been similar in the three groups. At 5 years, the ERACI III DES group had an increase in death either in comparison with BMS or CABG groups in the overall (1.84, 0.92–1.68, *p* = 0.08) and after adjusted population (2.53, 1.10,1.83, *p* = 0.03). Different adverse risk profiles among DES-treated patients cannot explain these differences, as multivariable statistical adjustment for baseline factors did not affect the results [30,31,32].

Therefore, in the ERACI III registry, patients treated with DES had a higher risk of overall death over the subsequent five years compared with patients treated with a BMS, despite a substantial reduction in the incidence of repeat coronary revascularization procedures.

However, at that time, we did not focus on cause-specific of death, and our major concern was restricted to the problem of late stent thrombosis, which was a major limitation of the first DES designs [33,34]. That is not more of a concern nowadays due to stent thrombosis, with the second and latest DES designs being similar or even lower than BMS, and it was not a safety limitation in any RCT or observational studies [1,2,35,36].

Of interest, if we make an indirect comparison between ISCHEMIA EXTENDED and the previous similar study COURAGE (Revascularization for Ischemic Ventricular Dysfunction) trial, a randomized comparison between PCI and OMT at 5 years of follow-up, in COURAGE there was no difference in either all-cause or cardiac and non-cardiac death between PCI or OMT [17]. Non-cardiac death was 5.4% vs. 6.1% with PCI and OMT, respectively, *p* = 0.59. In COURAGE, over 97% of patients included in the PCI group had been treated with BMS design.

## 5. The Hypothesis of High Non-Cardiac Death Incidence

Prolonged DAPT therapy may cause more bleeding events, and they have been linked with tumors [37]; however, we can speculate that if they caused more bleeding, they also would have early detection of the malignancies that would drive to a prompt and better treatment and that does not occur here. Furthermore, recently in a large RCT, clopidogrel monotherapy was associated with a lower risk in bleeding endpoints with no significant difference in the risk of all-cause death, compared to aspirin monotherapy [38], supporting the efficacy and safety of extended treatment with clopidogrel beyond 12 months.

The comment about these findings could be related to more radiation exposure due to repeat TVR or non-invasive diagnosis procedures in the invasive group not sustained by PCI history.

In the pre-DES era, repeat TVR was much more common due to a higher rate of ISR, and with older XR machines than currently, the presence of high non-cardiac death at that time was never reported [17,18,19,20,21].

Finally, the timeframe between radiation and tumors would need a more prolonged follow-up than the ISCHEMIA trial has had [7].

Endothelial dysfunction and impaired vessel motion after DES implantation were present in all designs, including complete biodegradable platforms [39,40], suggesting that the local action of the immunosuppressive drug might be the main reason for it, and endothelial dysfunction has been linked with a high incidence of solid tumors [41]. Furthermore, immunosuppressive agents, such as mTOR inhibitors, sirolimus, and everolimus, impair the immunosurveillance of neoplastic cells and have been associated with malignancies in kidney transplant patients [42].

Could endothelial dysfunction be more pronounced with multiple DES implantation?

We do not know the answer; however, since ISR is similar regardless of DES length, nowadays, we are seeing more and more stents implanted per patient in almost all PCI trials [43].

The observation in ISCHEMIA EXTEND that the non-cardiac cause of death is greater in patients with multiple vessel disease is going in such direction. Were patients dying for non-cardiac causes or cancer had more DES implanted or larger DES length? [7,15]

The comment that considered this is happening by chance is weak; this finding was not restricted to a single trial; that observation included 9903 patients randomized and treated in the last 6 years [3,4,5,6,7] (Table 1), and it would be very hard to believe that is happening by chance or a “false positive finding”.

Incidence of cardiac and non-cardiac death of RCT with DES has also been described in Figure 1 and Figure 2; of interest, non-cardiac death was higher in all studies, including those with mortality equipoise between PCI and CABG such as PRECOMBAT (Premier of Randomized Comparison of Bypass Surgery versus Angioplasty Using Sirolimus-Eluting Stent in Patients with Left Main Coronary Artery Disease) and NOBLE (Nordic-Baltic-British Left Main Revascularization Study) trials [44,45].

Pooled data from these 6 trials showed a trend to low cardiac death vs. the comparator *p* = 0.08 in favor (Figure 1) but a significant increase of non-cardiac death in the DES vs. comparator arm (5.5% vs. 3.8%, respectively, *p* = 0.000018) in favor to comparator Figure 2.

National regulatory agencies and scientific societies, together with principal investigators of those trials, should be involved in analyzing the clinical data of those patients having non-cardiac death or cancer.

Performing patient-level meta-analysis from RCT at long-term follow-up with similar comparators, DES vs. CABG, DES vs. OMT, or DES and CABG vs. OMT, we may also obtain useful information. Long-term follow-ups on ongoing studies such as FAME 3 (Fractional flow reserve versus angiography for the guidance of PCI in patients with multivessel disease) could add valuable information [43]. As was described above, a large trial-level meta-analysis published by Gaudino et al. [16] previously highlighted the differences in cause-specific of death among BMS, DES, and CABG trials; however, such findings were not incorporated in current clinical guidelines [46].

The use of drug-coated balloons have been used as an alternative to DES in some lesions subsets and could be an option to DES technology, particularly in small vessels or bifurcations stenosis [47,48]; however, the small sample size of RCT [49,50,51], as well as high late mortality, observed with these devices in peripheral arteries angioplasty [52] could be a limitation together with the potential concern to develop endothelial dysfunction due to local elution of immunosuppressive drug [41].

We can argue that the reduction of cardiac death counterbalances a high incidence of non-cardiac death, and maybe we would accept this fact. However, that is not always the same in different clinical scenarios as when we compared DES with CABG, where cardiac and non-cardiac death was lower in the last one [16]. Furthermore, in the pre-DES era, no differences in cardiac and non-cardiac death were reported in any meta-analysis from RCT [18,19,20], and that included the observation of overall mortality advantage in young patients treated with PCI [20]. Furthermore, no differences in cardiac and non-cardiac death were observed in the COURAGE trial, with more than 97% of BMS utilization [17]. All of those suggest the absence of any sign of non-cardiac compromise to late survival [20].

Lastly, findings of thin DES stent struts in the BIOSCIENCE trial were striking and suggested that the latest DES designs, despite their design improvements which translate into low target lesion revascularization, also hinted at safety concerns, as was shown by the high incidence of overall non-cardiac and cancer death [3].

## 6. Conclusions

The high incidence of non-cardiac death in DES trials is not new and was not seen with PCI in the pre-DES era. Nowadays, DES is used as the default strategy in all clinical and anatomical PCI scenarios all over the world, following current guidelines of major cardiology and interventional societies [46].

Considering these new findings, the universal use of DES could be tailored according to patient age, life expectancy, and anatomic complexity, such as LMCA [4], bifurcations [52], and chronic total coronary closure [53] until more information arrives, and we can discard if DES biology is related to this. It sounds rational to use DES in elderly patients, patients with complex coronary anatomy, with short life expectancy, or with previous or planned percutaneous aortic valve implantation where access to the coronary tree may be challenged [54]. In all these scenarios, DES use will be fully justified. However, it may be controversial if it was used in a young patient with no LMCA or left anterior descending artery compromise, where in several studies, no mortality benefit was demonstrated [1,2].

The topic discussed in this revision seems not resolved with new DES designs [3,12,40].

For now, it sounds mandatory that the cause-specific of death shall be monitored in the long term in all ongoing DES clinical trials.

As interventional cardiologists, we know that some of our colleagues may not agree with this viewpoint; however, patients’ safety comes first, and it is time to push all those involved in the PCI scenario to find out prompt answers.

## 7. Summary and Perspectives

The introduction of DES significantly improved the efficacy and safety of PCI, particularly in a high-risk group of patients, in comparison with old bare metal stent designs.

However, the gap between PCI with its competitors CABG and/or optimal medical treatment alone was not reduced.

In this revision, we highlighted that in recent years landmark randomized studies reported at mid- and long-term follow-ups a high incidence of non-cardiac death, cancer incidence, or both in the DES-treated group of patients.

Of note, non-cardiac death was more divergent between DES vs. the comparator at extended follow-up in favor of the last one.

High cancer occurrence occurred despite equal baseline prevalence of cancer among groups.

In one trial, cancer incidence was five times greater in the DES-treated group.

The reasons for these findings are now unexplained, although we can discard that DES biology could be involved in it.

Until all these issues are resolved, we propose that DES implantation should be tailored according to patient age, life expectancy, and lesion complexity.

## Figures and Tables

**Figure 1 diagnostics-13-01321-f001:**
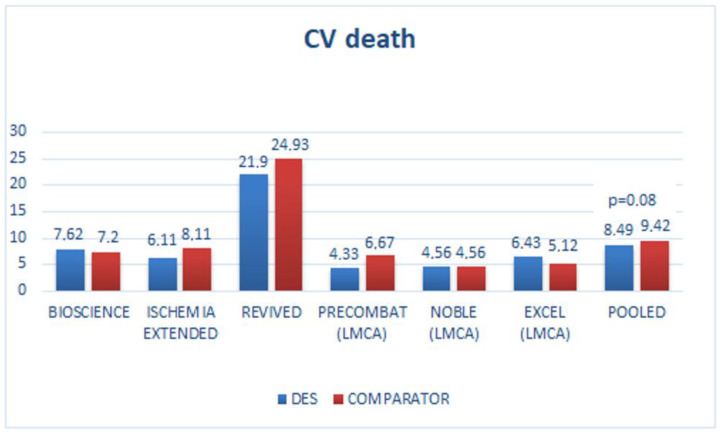
Randomized clinical trials with drug-eluting stents (DES) and cardiovascular death (CV-death). The last bar is the average of trial results. No significant differences in cardiac death (*p* = 0.08 in favor of DES). BIOSCIENCE: randomized third-generation DES vs. second-generation DES; ISCHEMIA Extended: randomized invasive strategy (DES in 74%) vs. optimal medical treatment (OMT); REVIVED: randomized DES vs. OMT; PRECOMBAT: randomized DES vs. CABG in left main coronary artery (LMCA); NOBLE: randomized DES vs. CABG in LMCA; EXCEL: randomized DES vs. CABG in LMCA.

**Figure 2 diagnostics-13-01321-f002:**
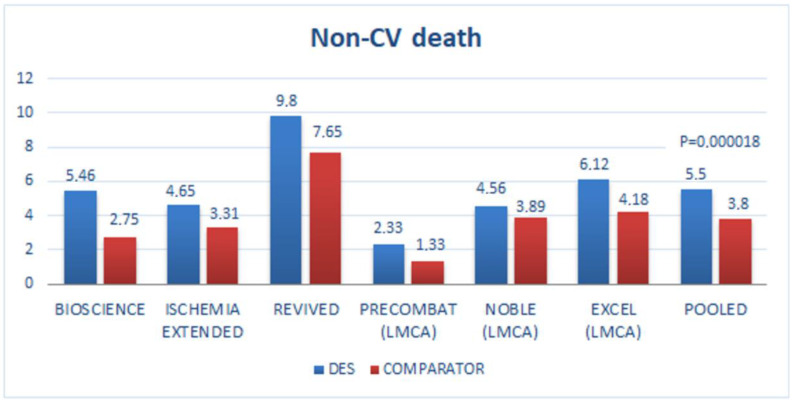
Randomized clinical trials with DES, non-cardiovascular death (non-CV death at long-term follow-up.) The last bar is the average of trial results. Significant differences in non-CV death (*p* = 0.000018 in favor of comparator). Acronyms, abbreviations, and trial designs are described in Figure 1.

**Table 1 diagnostics-13-01321-t001:** Randomized studies and meta-analyses with DES report a high incidence of NCD and/or cancer.

	Design	Comparator	Publication Year	Reference
Bioscience	RCT	DES3 vs. DES2	September 2018	[3]
EXCEL	RCT	DES2 vs. CABG	November 2019	[4]
ISCHEMIA	RCT	Invasive (74% DES2/26% CABG) vs. OMT	April 2020	[5]
REVIVED	RCT	DES2/3 vs. OMT	October 2022	[6]
ISCHEMIA EXTENDED	RCT	Invasive (74% DES2/26% CABG) vs. OMT	November 2022	[7]
Nordmann et al.	Meta-analysis	DES1 vs. BMS	December 2006	[10]
Gaudino et al.	Meta-analysis	BMS vs. CABG DES1 vs. CABG DES2 vs. CABG	December 2020	[8]

NCD: non-cardiac death, RCT: randomized clinical trials, DES3: third-generation drug-eluting stent, DES2: second-generation drug-eluting stent, CABG: coronary artery bypass surgery, OMT: optimal medical treatment, DES1: first-generation drug-eluting stent.

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
