# Peer review of "High Non-Cardiac Death Incidence Should Be a Limitation of Drug-Eluting Stents Implantation? Insights from Recent Randomized Data"

_diagnostics, 2023, doi:10.3390/diagnostics13071321_

Round 1

Reviewer 1 Report

The authors present the review article highlighting the outcomes after drug eluting stent implantation (DES) in comparison with other treatment strategies. Based on several randomized clinical trials they report the higher non-cardiac death incidence after DES compared to comparator arm. They question wether patients benefit from DES strategy. My main concern about the reported data that aiming at better  prognosis for patients we should focus on all-cause mortality as the higher risk of non-cardiovascular death could be balanced by better survival due to reduced cardiovascular death.

Major comments:

1) To my opinion it is worth to add to the Table the number of patients in each trial, the short description of patient groups, the end-points and the length of follow up.

2) The authors did not mention in their review such alternative strategy as drug-eluting balloon angioplasty though there are a number of studies comparing these two strategies.

3) The authors present BIOSCIENCE trial in which DES2 is compared to DES3. It is a little bit strange as the main focus of the review is to compare DES as a method of treatment to alternative methods.

Minor comments:

1) Please, use the same order of studies in the Table and in the Figures. It would be easier to get the information.

2) Please, check the abbreviation under the Table (BMS is missed).

3) Please, check the English grammar.

Author Response

Respond to Reviewers

Respond Reviewer 1

We thank the reviewer for his comments and advice about our manuscript.

We now extensively revised and rewrite the manuscript.

1-Reviewer Comment: General Comment: “. The authors present the review article highlighting the outcomes after drug eluting stent implantation (DES) in comparison with other treatment strategies. Based on several randomized clinical trials they report the higher non-cardiac death incidence after DES compared to comparator arm. They question wether patients benefit from DES strategy. My main concern about the reported data that aiming at better  prognosis for patients we should focus on all-cause mortality as the higher risk of non-cardiovascular

Respond: We have rewritten the manuscript and now we described in each trial who is reporting high non-cardiac death and/or cancer incidence.

On page 6 last paragraph of this new version now we added: “In summary in these RCTs briefly described above, BIOSCIENCE, EXCEL, and ISCHEMIA EXTENDED showed a significantly high incidence of non-cardiac death.

In BIOSCIENCE and ISCHEMIA EXTENDED non-cardiac death was driven by a high incidence of cancer death whereas in REVIVED with a lower length of follow-up than the other trials, a significantly high incidence of cancer occurrence was seen in the DES arm.

Of interest, this high cancer incidence in the DES group of patients observed in REVIVED would predict high late mortality in the DES-treated arm and a lack of benefit in the overall mortality rate of this trial.

2-Reviewer Major Comment: “ To my opinion it is worth to add to the Table the number of patients in each trial, the short description of patient groups, the end-points and the length of follow up.”

Respond: We modified the Table following the reviewer suggested.

3- Reviewer Comment:  “The authors did not mention in their review such alternative strategy as drug-eluting balloon angioplasty though there are a number of studies comparing these two strategies.”

Respond: We now added a large paragraph on page 10 last paragraph and also references: The use of drug-coated balloons have been used as an alternative to DES in some lesions subsets and could be an option to DES technology, particularly in small vessels or bifurcations stenosis (47-48), however, the small sample size of RCT (49-51) as well to high late mortality observed with these devices in peripheral arteries angioplasty(52) could be a limitation together with the potential concern to develop endothelial dysfunction due to local elution of the immunosuppressive drug.

4- Reviewer Comment: “The authors present BIOSCIENCE trial in which DES2 is compared to DES3. It is a little bit strange as the main focus of the review is to compare DES as a method of treatment to alternative methods.”

Respond: This is a new DES design with thin struts stents that are associated with a reduction of target lesion revascularization compared to second-generation DES mainly XIENCE stents.

However, is striking the differences in non-cardiac death and cancer death associated with this device, we added a paragraph on page 11 first paragraph of the new version of the manuscript :

Lastly, findings of thin DES stent struts in the BIOSCIENCE trial were striking and suggested that the latest DES designs, despite their design improvements which translate into low target lesion revascularization, also hinted at safety concerns as was shown by the high incidence of overall, non-cardiac and cancer death (3).

And also, in the fourth paragraph of the conclusions: The topic discussed in this revision seems not resolved with new DES designs (3,12,40).

We thank all the minor comments of the reviewer, and we resolve all of them following his suggestions.

Finally, the entire text has been reviewed extensively by a native English corrector.

Reviewer 2 Report

The article has challenged to elaborate on a difficult issue regarding the mode of death in patients receiving coronary revascularization. It concluded that death of non-cardiac cause in patients with DES implantation has been known since the emergence of DES, and the use of DES should be individualized according to patient's characteristics and anatomies of coronary arteries. This opinion suggesting possible Achilles' heel of the use of DES has a substatial impact on desicion-making on therapeutic choice in patients with stabl CAD. The article style is a narrative review rather than a systematic one.

However, I have noticed severeal concerns on the paper.

Major comments

1) The whole style and context are not neatly structured. Please consider resetting the title and main topic of the each paragraph, and reorder sentences to facilitate reader's understanding.

2)  It should be clearly shown as to whether the cause of death referred is non-cardiac etiologily or specifically cancer. This point is ambigous in some sentences, which makes discussion obsecure. The speculated non-cardiac mechanism under which patients died after DES implantation should be discussed separately on cancer and other potential causes.

3) As mentioned in the Conclusions, the effect of age should be taken into account for referring to non-cardiac death.

Minor comments,

1) Multiple typo and grammatical errors are found.     

Author Response

Comments Reviewer 2

We thank the reviewer’s comments and advice about our manuscript.

We have extensively reviewed and rewrote the manuscript accorded his comments.

  • Major Comment:” The whole style and context are not neatly structured. Please consider resetting the title and main topic of the each paragraph, and reorder sentences to facilitate readers understanding”

Respond: We rewrote the main topic of each paragraph. In the description of each of the trials, we add a subheading:   the Trial list, 2nd paragraph of the third page.

In the Trial list, we described the endpoints and major concerns of each of the trials.

2)  Major Comment: “It should be clearly shown as to whether the cause of death referred is non-cardiac etiologily or specifically cancer. This point is ambigous in some sentences, which makes discussion obsecure. The speculated non-cardiac mechanism under which patients died after DES implantation should be discussed separately on cancer and other potential causes”.

Respond: We change the last subheading title before the Conclusions section, and now we rename it as Potential Hypothesis of High Non-cardiac Death Incidence.

In this section, we largely discuss potential reasons for non-cardiac death including cancer, on pages 9 and 10 of the new version of the manuscript.

We also split out causes of death in each trial: and now we described who is reporting high non-cardiac death and/or cancer incidence.

On page 6 last paragraph of this new version now we added: “In summary in these RCTs briefly described above, BIOSCIENCE, EXCEL, and ISCHEMIA EXTENDED showed a significantly high incidence of non-cardiac death.

In BIOSCIENCE and ISCHEMIA EXTENDED non-cardiac death was driven by a high incidence of cancer death whereas in REVIVED with a lower length of follow-up than the other trials, a significantly high incidence of cancer occurrence was seen in the DES arm.

Of interest, this high cancer incidence in the DES group of patients observed in REVIVED would predict high late mortality in the DES-treated arm and a lack of benefit in the overall mortality rate of this trial.

3-Reviewer Major Comment: “As mentioned in the Conclusions, the effect of age should be taken into account for referring to non-cardiac death.”

Respond: We extend our thoughts and rationality on the DES use accorded age, life expectancy, and lesion complexity in Conclusions, we added a large paragraph on it and references, page 11  3rd. and 4th paragraph:

“Considering these new findings, the universal use of DES could be tailored according to patient age, life expectancy, and anatomic complexity such as LMCA (4,), bifurcations (52), chronic total coronary closure (53) until more information arrives and we can discard if DES biology is related to this.

Sounds rational to use DES in elderly patients, patients with complex coronary anatomy, with short life expectancy, or with previous or planned percutaneous aortic valve implantation where access to the coronary tree may be challenged (54). In all these scenarios DES use will be fully justified. However, may be controversial if it was used in a young patient with no LMCA or left anterior descending artery compromise where in several studies no mortality benefit was demonstrated (1-2).

The topic discussed in this revision seems not resolved with new DES designs.”

Reviewer Minor Comments about English style.

Respond: The manuscript was extensively revised by a native English

Round 2

Reviewer 1 Report

I would like to thank the authors for their response. The corrections have improved the manuscript. I do not have further comments. 

Reviewer 2 Report

The authors have replied to my first review in a point-by-point fashion. Overall, the revised version is satisfactorily corrected.